# Evaluation of Dosing Guidelines for Gentamicin in Neonates and Children

**DOI:** 10.3390/antibiotics12050810

**Published:** 2023-04-25

**Authors:** Esther M. Hollander, Eline L. van Tuinen, Elisabeth H. Schölvinck, Klasien A. Bergman, Arno R. Bourgonje, Valentina Gracchi, Martin C. J. Kneyber, Daan J. Touw, Paola Mian

**Affiliations:** 1Department of Clinical Pharmacy and Pharmacology, University Medical Center Groningen, University of Groningen, Hanzeplein 1, 9713 GZ Groningen, The Netherlands; 2Department of Pediatric Infectious Diseases, Beatrix Children’s Hospital, University Medical Center Groningen, University of Groningen, Hanzeplein 1, 9713 GZ Groningen, The Netherlands; 3Division of Neonatology, Department of Pediatrics, Beatrix Children’s Hospital, University Medical Center Groningen, University of Groningen, Hanzeplein 1, 9713 GZ Groningen, The Netherlands; 4Department of Gastroenterology and Hepatology, University Medical Center Groningen, University of Groningen, 9713 GZ Groningen, The Netherlands; a.r.bourgonje@umcg.nl; 5Division of Pediatric Nephrology, Beatrix Children’s Hospital, University Medical Center Groningen, University of Groningen, Hanzeplein 1, 9713 GZ Groningen, The Netherlands; 6Division of Peadiatric Critical Care Medicine, Department of Paediatrics, Beatrix Children’s Hospital Groningen, University Medical Center Groningen, University of Groningen Hanzeplein 1, 9713 GZ Groningen, The Netherlands; 7Department of Pharmaceutical Analysis, Groningen Research Institute for Pharmacy, University of Groningen, Hanzeplein 1, 9713 GZ Groningen, The Netherlands

**Keywords:** gentamicin, pharmacokinetics, neonates, children, dosage regimen

## Abstract

Although aminoglycosides are frequently prescribed to neonates and children, the ability to reach effective and safe target concentrations with the currently used dosing regimens remains unclear. This study aims to evaluate the target attainment of the currently used dosing regimens for gentamicin in neonates and children. We conducted a retrospective single-center cohort study in neonates and children receiving gentamicin between January 2019 and July 2022, in the Beatrix Children’s Hospital. The first gentamicin concentration used for therapeutic drug monitoring was collected for each patient, in conjunction with information on dosing and clinical status. Target trough concentrations were ≤1 mg/L for neonates and ≤0.5 mg/L for children. Target peak concentrations were 8–12 mg/L for neonates and 15–20 mg/L for children. In total, 658 patients were included (335 neonates and 323 children). Trough concentrations were outside the target range in 46.2% and 9.9% of neonates and children, respectively. Peak concentrations were outside the target range in 46.0% and 68.7% of neonates and children, respectively. In children, higher creatinine concentrations were associated with higher gentamicin trough concentrations. This study corroborates earlier observational studies showing that, with a standard dose, drug concentration targets were met in only approximately 50% of the cases. Our findings show that additional parameters are needed to improve target attainment.

## 1. Introduction

Gentamicin is an aminoglycoside frequently prescribed in neonates and children for the treatment of severe bacterial infections or for prophylaxis of high-risk infections [1]. It is effective against a wide range of bacteria, including most Gram-negative bacteria and Staphylococcus aureus [2]. Like other aminoglycosides, gentamicin has a narrow therapeutic window. Its antibacterial effect is concentration dependent and the ratio of peak concentration (Cpeak) to the minimal inhibitory concentration (MIC) has been proposed as a pharmacokinetic/pharmacodynamic (PK/PD) predictor of efficacy [3,4]. Overexposure to gentamicin has been associated with nephrotoxicity and ototoxicity [5,6]. 

The pharmacokinetic parameters of gentamicin can vary considerably in neonates and children. Clearance ranges from 0.49 to 6.3 L/h/70 kg and 5.6 to 9.1 L/h/70 kg, and volume of distribution ranges from 26.6 to 63.7 L/70 kg and 17.5 to 24.5 L/70 kg, respectively [7]. These variabilities in pharmacokinetics are due to changes in body composition and organ maturation and development, especially in children younger than 2 years of age [8]. In healthy neonates, the half-life can decrease by more than 50% over the first 7 to 10 days after birth [9]. Due to this pharmacokinetic variability in neonates and children, in combination with a narrow therapeutic window, appropriate dosing of gentamicin in these populations is challenging and therapeutic drug monitoring (TDM) is recommended, to individualize the dosage and achieve target concentrations in each patient [10,11]. Target trough concentrations are mandatory for safety, with an advised trough of ≤1 mg/L for neonates and ≤0.5 mg/L for children. Target peak concentrations suggested for efficacy are 8–12 mg/L and 15–20 mg/L for neonates and children, respectively [5,12,13].

Once-daily administration of gentamicin is preferred over multiple times per day, to optimize effectiveness and reduce toxicity [14,15,16,17]. For neonates and children, the literature supports initial dosing ranges from 3 to 7.5 mg/kg and 4 to 10 mg/kg, respectively, usually given once daily, with subsequent doses determined by TDM [18,19,20,21,22]. The Dutch Pediatric Formulary (DPF) aims to unify prescribing for neonates and children, by providing best evidence-based dosing recommendations, and is currently the national standard for pediatric pharmacotherapy in the Netherlands [23,24]. In the Beatrix Children’s Hospital (Beatrix KinderZiekenhuis, BKZ), dosing recommendations of the DPF are followed for children above 1 month of age, while they are not for neonates (Table 1). The latter is based on clinical practice. The advised dose of gentamicin in children aged 12 to 18 years was modified and posted on the DPF’s open-access internet platform (https://www.kinderformularium.nl; accessed on 20 January 2021), as shown in Table 1. This dose adjustment is based on pharmacokinetic (PK) data from several population PK studies [22,25,26]. However, the ability to reach PK/PD targets of efficacy and safety with these dosing regimens is unclear. Therefore, this study aims to evaluate the target attainment of the currently used dosing guidelines for gentamicin in neonates and children.

## 2. Results

### 2.1. Patient Characteristics

The screening procedure resulted in the selection of 864 patients (for a total of 1002 gentamicin treatment cycles). Of these, 206 patients (295 cycles) were excluded, mainly because gentamicin treatment was not started in BKZ or because patients received gentamicin in the two weeks prior to the new treatment cycle. This resulted in 658 patients for data analysis, amounting to a total of 707 gentamicin treatment cycles (Figure 1). An overview of the clinical characteristics of each cohort is shown in Table 2.

### 2.2. Patient Cohorts

#### 2.2.1. Neonates

In total, 335 patients (349 cycles) were included in this cohort, with the majority (58.2%) being term neonates (GA > 37 weeks) (Table 2). Median gentamicin trough and peak concentrations were both within the target range (0.92 and 11.14 mg/L, respectively), but showed significant interindividual variation, ranging from 0.00 to 4.07 and 5.50 to 30.87 mg/L, respectively (Table 2, Figure 2A). In 37 cycles, the initial starting dose deviated from the advised dose by the BKZ or DPF (e.g., improper dose or improper frequency), leaving 318 gentamicin cycles for target attainment analysis (312 trough concentrations and 274 peak concentrations).

#### 2.2.2. Children

In total, 323 patients (358 cycles) were included in the children cohort, with a median age of 2.21 years, ranging from 34 days to 17.9 years (Table 2). The median gentamicin trough and peak concentrations were 0.04 and 14.77 mg/L, respectively. In 11 cycles, the initial starting dose deviated from the advised dose by the DPF, leaving 347 cycles for analysis (334 trough concentrations and 317 peak concentrations). As of January 2021, 38 children aged 12 to 18 years received gentamicin, of which 27 (71.1%) received 5 mg/kg/day and 11 (28.9%) received 7 mg/kg/day. Before January 2021, all children aged 12 to 18 years (*n* = 34) received 7 mg/kg/day, except for one patient receiving 5 mg/kg/day.

### 2.3. Target Attainment

The percentages of target attainment, arranged by gentamicin dose and age group, are presented in Table 3, with a visual representation in Figure 2. All cohorts and subgroups showed significant interindividual variation, even though all patients within a subgroup received a similar daily dose and dose interval.

#### 2.3.1. Neonates

Fifty-four percent of the neonates had therapeutic peak concentrations, while 10.6% and 35.4% of them had subtherapeutic and supratherapeutic concentrations, respectively (Figure 2A, Table 3). Of all neonates receiving 5 mg/kg, term neonates showed the worst safety target attainment (34.9%). Within the subgroup of term neonates, we found a higher percentage of safety target attainment in neonates receiving 4 mg/kg compared to neonates receiving 5 mg/kg (85.3% and 34.9%, respectively), while target attainment of peak concentrations was similar (54.5% and 54.6%, respectively). Both within the subgroups of <32 weeks GA and 32–37 weeks GA, we see a difference in target attainment between neonates <7 days PNA and ≥7 days PNA. Neonates <7 days PNA showed a higher percentage of supratherapeutic concentrations. Target attainment of therapeutic concentrations was higher within neonates ≥7 days PNA. Furthermore, the attainment of safe trough concentrations was higher for neonates ≥7 days PNA. In addition, linear mixed model analysis was performed to confirm the observed differences in gentamicin peak and trough concentrations among the age subgroups (fixed effect) while allowing the presence of multiple concentration measurements for some neonates (random effect for individual subjects). When grouping all neonates together per age subgroup (<32 weeks GA, 32–37 weeks GA, and term neonates), neonates at term showed significantly lower peak concentrations but higher trough concentrations compared to neonates <32 weeks GA (*p* = 0.010 and *p* < 0.001, respectively). When studying differences between <7 and ≥7 days PNA, peak concentrations were significantly higher in neonates <7 days PNA compared to neonates ≥7 days PNA in both the <32 weeks GA and 32–37 subgroups (*p* < 0.001 and *p* = 0.001, respectively). Similarly, trough concentrations were higher in neonates <7 days PNA compared to neonates ≥7 days PNA in both the <32 weeks GA and 32–37 weeks subgroups (*p* = 0.005 and *p* < 0.001, respectively).

#### 2.3.2. Children

Gentamicin peak concentrations were predominantly subtherapeutic (51.7%), and the majority of trough concentrations (90.1%) were below the toxicity threshold of ≤0.5 mg/L (Figure 2B, Table 3). Within the subgroup of children aged 12 to 18 years, we found a higher percentage of safety target attainment in children receiving 5 mg/kg compared to children receiving 7 mg/kg (100.0% and 86.0%, respectively), while target attainment of peak concentrations was similar (25.9% and 22.2%, respectively). We saw a shift from a high percentage of subtherapeutic concentrations in children receiving 5 mg/kg (63.0%) to supratherapeutic concentrations in children receiving 7 mg/kg (47.2%) (Table 3). 

### 2.4. Correlation with Creatinine Concentration in Children above 1 Month of Age

As creatinine concentrations are not routinely checked during regular care in our center, these data were frequently missing: in, respectively, 28% and 34% of all cycles, no creatinine concentration was determined within 24 h of the start of gentamicin or within 24 h of the TDM sample. The relationship between creatinine at the start of treatment and trough concentration was statistically significant for gentamicin (Spearman’s rho 0.365; *p* < 0.001; N = 258) (Figure 3A). Creatinine concentrations on the day of TDM and gentamicin trough concentrations were also correlated (Spearman’s rho 0.432; *p* < 0.001; N = 236) (Figure 3B).

## 3. Discussion

In this retrospective cohort study, we investigated the target attainment of gentamicin in neonates and children. For neonates, median gentamicin concentrations fell within the nontoxic, therapeutic range. However, due to the large interindividual variability, a significant proportion of neonates showed toxic trough levels and/or supra- and subtherapeutic peak levels, with term neonates receiving 5 mg/kg every 24 h showing the most suboptimal safety target attainment. In preterm neonates, we observe a difference in target attainment between neonates aged <7 days and neonates aged ≥7 days. Most of the children showed safe trough levels but also subtherapeutic peak levels. Efficacy target attainment was the lowest in children >12 years receiving 7 mg/kg.

### 3.1. Preterm Neonates (<32 Weeks GA and 32–37 Weeks GA) 

Target attainment was similar for neonates <32 weeks GA and neonates 32–37 weeks GA. However, we did observe a difference in target attainment between preterm neonates aged <7 days and preterm neonates aged ≥7 days. Our results show higher efficacy and safety target attainment in neonates aged ≥7 days compared to neonates aged <7 days, while receiving the same dose of 5 mg/kg. Van Donge et al. also observed that neonates aged <7 days showed toxic trough concentrations more frequently than neonates aged ≥7 days with a dosing regimen of 7.5 mg/kg/36 h [27]. Hartman et al., who investigated dosing regimens and pharmacodynamic targets in a comparable population (Appendix A Table A1), found a similar efficacy target attainment for this group as we found in our study. In Appendix A Table A1, an overview of the observational PKPD studies investigating gentamicin dosing regimens is provided. In contrast to our results, most studies found that peak concentrations were mostly subtherapeutic, whereas ours were mostly supratherapeutic. Moreover, we found a better safety target attainment, with safe trough concentrations in 100% and 89% of neonates <32 weeks GA and 32–37 weeks GA, respectively [28]. Veltkamp et al. expected a dosing regimen of 5 mg/kg/36 h in neonates <35 weeks GA and <7 days PNA to result in the least number of dose adjustments, based on pharmacokinetic simulations [29]. Sum et al. confirmed this expectation and found that this dosing regimen led to the attainment of therapeutic peak concentrations and safe trough concentrations in 50% and 91% of neonates, respectively [30]. Considering that we found a significant proportion of supratherapeutic peak concentrations and toxic trough concentrations in preterm neonates <7 days, we do not recommend the dosing regimen of the BKZ. Instead, we suggest considering a dosing advice of 4 mg/kg/24 h. 

For preterm neonates ≥7 days, the dosing regimen of the BKZ (5 mg/kg) differs from the dosing regimen recommended by the DPF (4 mg/kg). We found the highest safety and efficacy target attainment in preterm neonates ≥7 days. In contrast to our results, Hartman et al. found the lowest target attainment in neonates aged ≥7 days, with 100% of peak concentrations being subtherapeutic [28]. As Hartman et al. considered target attainment of the dosing regimen by the DPF, these findings suggest that the dosing regimen of the BKZ leads to a higher efficacy in preterm neonates aged ≥7 days. Therefore, we recommend implementing the 5 mg/kg/24 h dosage for preterm neonates aged ≥7 days. 

### 3.2. Term Neonates (>37 Weeks)

For term neonates, we do not recommend the dosing regimen of the BKZ, because of the high frequency of toxic trough concentrations. The dosing regimen of the DPF shows a higher attainment of safe trough concentrations compared to the dosing regimen of the BKZ, while efficacy target attainment was similar with both dosing regimens. This implies that the currently used dosing regimen in the BKZ seems to overestimate the clearance of term neonates. These findings suggest that the dosing regimen of the DPF leads to a more desirable outcome. However, Hartman et al. found a low efficacy attainment of gentamicin with the dosing regimen recommended by the DPF, with only 13% of term neonates showing therapeutic concentrations (Appendix A Table A1). Similar to our results, they found a relatively high safety attainment of gentamicin with the dosing regimen recommended by the DPF, with 71% of term neonates showing safe concentrations [28]. Sum et al. found that a gentamicin dosing regimen of 5 mg/kg/36 h led to the attainment of therapeutic peak concentrations and safe trough concentrations in 47% and 98% of neonates >35 weeks, respectively [30]. Therefore, we propose to use the DPF dosing (4 mg/kg/24 h), despite suboptimal exposure, as seen in our study and those from Hartman et al. [28]. 

### 3.3. Children 1–12 Years

For children <12 years, the current gentamicin dosing guideline of the DPF seems to underestimate the overall volume of distribution (Vd) for children aged from 1 month to 12 years, as is shown by a large proportion of subtherapeutic peak concentrations. Therefore, the DPF should re-evaluate this dosing guideline by considering a higher initial dose, especially for infants < 1 year of age, to reach the desired peak levels. Furthermore, in line with several studies performed on gentamicin PK in neonates, we see large interpatient variability in gentamicin pharmacokinetics in children [28,29,30]. To the best of our knowledge, no observational PKPD studies providing gentamicin dosing regimens have been conducted in children. Therefore, that information could not be included in Appendix A Table A1. Given that all children <12 years of age in our study received the same dose, the large variability between patients was to be expected. From a pharmacological point of view, in fact, children of a younger age receiving the same dose as children of an older age can be expected to achieve lower peak concentrations of gentamicin, as Vd decreases with age, from about 0.39 L/kg in children from 0 to 20 months to about 0.25 l/kg in young adults [21,31]. This aligns with studies suggesting younger children may need higher starting doses than the standard dose of 7 mg/kg [20,22,26]. On the other hand, gentamicin clearance (CL) is determined by the developmental stage of the renal function, which is fully matured at the age of 1–2 years [32,33]. Therefore, it may be of added value to develop dosing guidelines for different age groups within children <12 years based on PK differences (e.g., infants (2–12 months), toddlers (1–2 years), and children (5–10 years)) [7].

### 3.4. Children 12–18 Years

For children >12 years, the DPF’s dosing regimen of gentamicin was altered in January 2021. We expected higher target attainment in children receiving the new advised dosage (5 mg/kg) compared to the old dosage (7 mg/kg). However, the large proportion of subtherapeutic and supratherapeutic peak concentrations in children >12 years receiving 5 mg/kg and 7 mg/kg, respectively, show that both dosing regimens are suboptimal. To our knowledge, no other studies have been published on external real-world validation of gentamicin dosing guidelines in children. 

As previously stated, several different gentamicin dosing regimens have been proposed by studies, ranging from 3 to 7.5 mg/kg for neonates and 4 to 10 mg/kg for children [20,21,29,30]. Beyond the differences in population, some of the differences in proposed dosing regimens are due to different target concentrations, primarily based on Cmax/MIC or AUC/MIC ratio [34]. In our study, target concentrations were based on clinical studies from the 1980s and 1990s, showing the Cmax/MIC ratio to be the PK/PD index that was primarily linked to clinical efficacy, with maximal efficacy at Cmax/MIC ≥ 8–10 and an MIC of 1 mg/L [35,36,37]. According to the European Committee on Antimicrobial Susceptibility Testing (EUCAST), gentamicin MIC breakpoints for Enterobacteriaceae are ≤2 mg/L, which leads to a target Cmax of 16–20 mg/L for gentamicin [38]. Since MICs of 1 mg/L have been increasingly observed, especially for pathogens encountered in early and late neonatal onset sepsis (Escheria coli, Pseudomonas and Klebsiella spp.), the target peak concentration for neonates was set at 8–12 mg/L [39,40]. In the literature, trough concentrations are indicated as <0.5–2 mg/L, with trough concentrations of 2 mg/L, possibly leading to a higher risk of toxicity [11]. Compared to children and adults, the risk of gentamicin toxicity in neonates is believed to be lower, due to a larger volume of distribution [41]. Therefore, target trough concentrations were set at <1 for neonates and <0.5 mg/L for children. However, nephrotoxicity and ototoxicity also occur in children when adequate trough concentrations are maintained, and an every-other-day regimen is used [28].

The relatively high percentages of children missing serum creatinine measurements are of concern given that renal dysfunction is acknowledged as an important factor warranting gentamicin dose adjustment. Determining creatinine levels should not be a problem, as it can be easily determined from the same blood sample taken for TDM. Since aminoglycosides are primarily cleared through renal elimination by glomerular filtration, it is not surprising that an elevated serum creatinine concentration was associated with delayed gentamicin clearance. Several other studies found the same results [31,42]. Serum creatinine concentrations correlated moderately well with gentamicin trough concentrations at both dosing and sampling, therefore, these concentrations could both be used for further optimization and individualization of dosing recommendations. Any interval extension can still be applied before giving the second dose. Other studies showed a correlation between creatinine concentrations and gentamicin concentrations in neonates [38,43,44,45,46]. Nevertheless, we chose not to analyze neonatal serum creatinine values in relation to gentamicin concentrations, because serum creatinine in the first weeks of life is an unreliable measure of GFR in neonates and reflects, specifically in the first days of life, maternal serum creatinine concentration [47,48]. Moreover, under physiological circumstances, preterm neonates may exhibit a transient increase in serum creatinine, reaching a peak value between the second and fourth day of life, probably due to passive reabsorption of filtered creatinine by immature tubules [49,50]. 

### 3.5. Recommendations

It can be concluded that there is a need to validate dosing advice in real-world settings after it has been published or adopted in (inter)national guidelines, as supra- and subtherapeutic peak gentamicin concentrations were frequent in neonates and children, respectively. For neonates, such studies have already been performed, but for children, data are still lacking. As mentioned, the observed interpatient variability in neonates and children is large, and is one of the main factors contributing to the difficulty of dosing in this heterogeneous group of patients. In this study, we investigated serum creatinine as a possible covariate in children, but other factors could also influence gentamicin concentrations. Several studies described covariates for their potential influence on gentamicin PK parameters: (gestational) age, (birth) weight, and admission unit [31,42,51,52]. These additional covariates could also be a significant source of variability that will need to be addressed in future research. If such variables cannot be identified, the best option is to widely adapt the early limited sampling concept. Early limited sampling reduces the burden of TDM by using a small number of appropriately timed samples and prevents prolonged periods with elevated trough concentrations. Furthermore, complicated dosing regimens can be avoided with this concept, since a standardized loading dose will be used. Future research should also focus on the external validation of pharmacokinetic models in daily clinical practice. 

### 3.6. Limitations

Our study had several limitations. First, this was a retrospective study using clinical data that were not specifically collected for research purposes, which resulted in a relatively high percentage of missing data. Second, we depended on the reports in the electronic patient files for the dose administration and blood sampling times, the accuracy of which is not precisely known. Another limitation was the fact that we defined gentamicin plasma concentrations that were obtained between 1.5 and 23, 36, or 47 h (depending on the dosing interval) after the start of administration as mid-concentrations. Mid-concentrations were all extrapolated to peak and trough concentrations, which may have led to some inaccuracy in calculated concentrations. Furthermore, we only included the attainment of target concentrations as a measure of efficacy and safety, as we did not take response on therapy or the occurrence of nephrotoxicity and ototoxicity into account. Therefore, we cannot state with certainty whether a treatment cycle was effective or safe. Additionally, it is daily practice to only determine gentamicin levels in a select group of neonates, to be precise, those with renal impairment and those who continue gentamicin treatment beyond 24–48 h. This means that we looked at relatively ill neonates in this study. Lastly, no distinction was made between the degree of illness. As critically ill pediatric patients are subject to even larger pharmacokinetic variability, due to pathophysiological changes affecting Vd and CL, gentamicin concentrations may be affected by these factors [53].

## 4. Materials and Methods

### 4.1. Study Design

We conducted a retrospective, single-center cohort study of neonates and children admitted to the BKZ between 1 January 2019 and 13 July 2022. Patients were identified using a query in the hospital’s electronic information system (EPIC software, Verona, WI, USA). Due to this study’s retrospective and observational nature, the need to provide informed consent was waived by the University Medical Center Groningen (UMCG) medical ethics committee (METc 2022/527).

### 4.2. Inclusion and Exclusion Criteria

All patients younger than 18 years of age, admitted to the BKZ and with at least one gentamicin concentration determined as part of TDM, were eligible for this study. The following definitions were used: preterm neonates: GA < 37 weeks, term neonates: >37 weeks GA–1 month, children: 1 month–12 years, adolescents 12–18 years. Patients were excluded in cases of (1) gentamicin prescription for the indication of endocarditis, as the recommended initial dose differs from the initial dose for severe infections or prophylaxis, (2) treatment was started in another hospital, (3) extracorporeal membrane oxygenation (ECMO) or any form of renal replacement therapy was required during gentamicin treatment, (4) therapeutic hypothermia for perinatal asphyxia, (5) gentamicin treatment in the two previous weeks, (6) no accurately timed dose administration or sample drawn for trough or peak concentration measurement as part of TDM or the sample was drawn during administration. If a patient had received intravenous gentamicin several times in the past three years, these treatment occasions were included as separate cycles if there were at least two weeks between the last dose of the previous cycle and the first dose of the new cycle, as stated in the fourth exclusion criterion. A cycle involves a treatment of at least 1 dose of gentamicin.

Per treatment cycle, we included only the first peak and trough concentration of gentamicin after therapy initiation and before TDM-guided alterations in dose or dose interval had occurred, as concentrations determined after TDM would not reflect the initial target attainment of the initial dose. We chose not to exclude patients whose initial dosage deviated from the recommended doses by the DPF or BKZ as long as they received it for severe infections or as prophylaxis, to truly reflect the use of gentamicin in current clinical practice. However, for the target attainment analysis, we only included patients with an initial dose as recommended by the DPF or BKZ.

### 4.3. Data Collection

Clinical data were collected retrospectively from electronic health records from 1 January 2019 to 13 July 2022. The collected data were basic demographic characteristics, admission data, gentamicin doses and their respective date and time of administration, gentamicin plasma concentrations and their respective date and time of sampling, and serum creatinine concentrations for children. For neonates, gestational age (GA) was also discriminated. We did not include serum creatinine concentrations for neonates, as these concentrations reflect the maternal creatinine concentration in the first couple of postnatal weeks [47]. Drug concentrations were analyzed using PETINIA (Particle-Enhanced Turbidimetric Inhibition Immuno-assay) (Architect c8000, Abbott Laboratories, Lake Bluf, IL, USA) for gentamicin with a lower and upper limit of quantification of 0.5 and 36 mg/L, respectively [54,55]. A gentamicin concentration was considered a peak concentration if the sample was drawn between 20 min and 1 h after the end of infusion, and a trough concentration if the sample was drawn at least 23, 35, or 47 h after the start of gentamicin administration, depending on the dose interval. If a mid-concentration (a sample drawn 1.5, 23, 35, or 47 h after the start of gentamicin) was determined, or when the collection time deviated from the defined times, a Bayesian one-compartment pharmacokinetic model (MwPharm++ version 2.21; Mediware, Prague, Czech Republic) was used to calculate peak and trough concentrations.

### 4.4. Data Analysis

Seven subcohorts were established within the two main cohorts (neonates and children) due to the differences in gentamicin dosing advice, namely: neonates <32 weeks GA and <7 days PNA, neonates <32 weeks GA and ≥7 days PNA, neonates 32–37 weeks GA and <7 days PNA, neonates 32–37 weeks GA and ≥7 days PNA, neonates >37 weeks GA, children aged one month to 12 years and children aged 12 to 18 years. Within these cohorts, peak concentrations were divided into three subgroups (subtherapeutic, therapeutic, and supratherapeutic), whereas trough concentrations were divided into two subgroups (safe and toxic), according to the target concentrations for efficacy and/or safety. Therapeutic targets for drug efficacy were considered as a gentamicin peak concentration of 8–12 mg/L for neonates and 15–20 mg/L for children. Target trough concentrations for gentamicin safety were <1 mg/L and <0.5 mg/L for neonates and children, respectively. Descriptive data are presented as median and interquartile ranges (IQR) and categorical data are presented as whole numbers or percentages. Attainment of the pharmacodynamic targets is presented as a percentage of cycles in each of the subgroups. 

The correlation between gentamicin trough concentrations and creatinine concentrations in children was assessed with Spearman’s rank correlation test, using SPSS Statistics version 28 (IBM, Armonk NY, USA). Serum creatinine concentrations within 24 h before TDM, were used as a covariate for glomerular filtration rate (GFR), as gentamicin is predominantly excreted by glomerular filtration. Subsequently, we tested the correlation of creatinine concentrations within 24 h before the start of treatment with gentamicin, to investigate whether creatinine could guide antibiotic dosing at treatment commencement.

## 5. Conclusions

This study shows the importance of external, real-world validation of dosing guidelines in pediatric populations, as both sub- and supratherapeutic concentrations of gentamicin were highly prevalent. Safety and efficacy target attainment was most suboptimal in term neonates receiving 5 mg/kg and children >12 years receiving 7 mg/kg. In addition, our data underline the necessity to describe the variables accounting for the interpatient variability. If such variables cannot be identified, the best option is to widely implement the concept of “early limited sampling”.

## Figures and Tables

**Figure 1 antibiotics-12-00810-f001:**
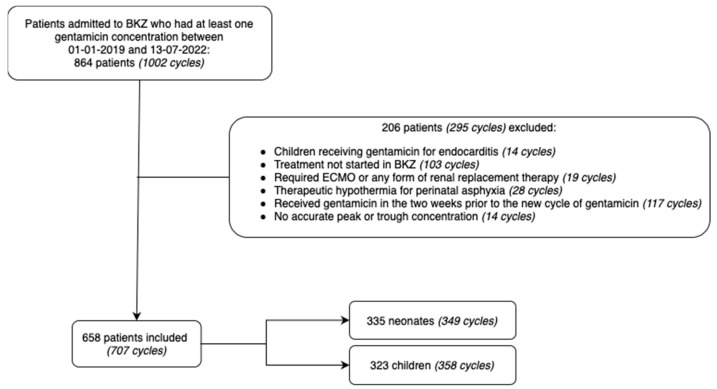
Flowchart indicating the total number of cycles, number of exclusions, reason for exclusion, total number of patients and gentamicin treatment cycles included, and stratification among the cohorts. BKZ: Beatrix Children’s Hospital, ECMO: extracorporeal membrane oxygenation.

**Figure 2 antibiotics-12-00810-f002:**
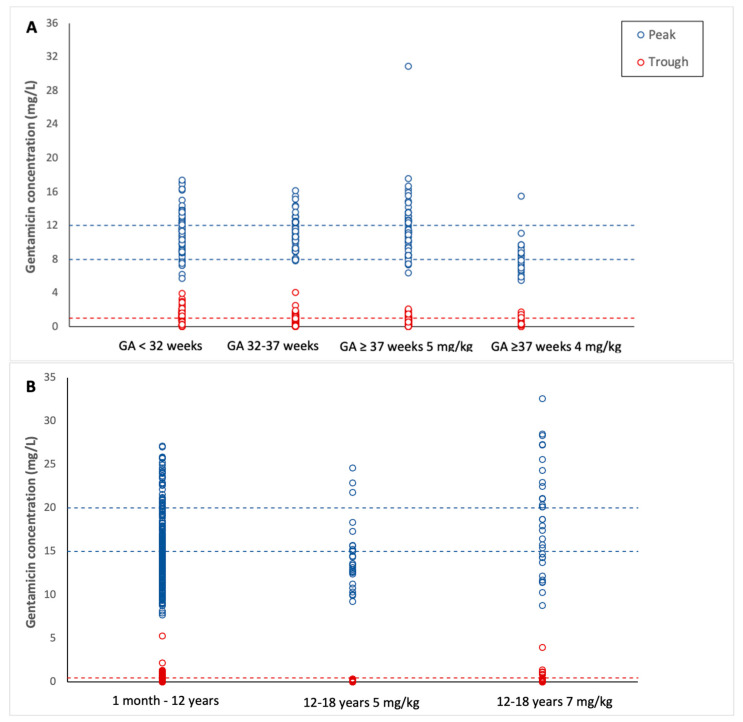
Concentrations of gentamicin in neonates and children. (**A**) Neonates, (**B**) children. Blue circles represent a single-patient peak concentration, and red circles represent single-patient gentamicin trough concentrations (as delineated in Table 3). Dashed lines indicate the target concentrations.

**Figure 3 antibiotics-12-00810-f003:**
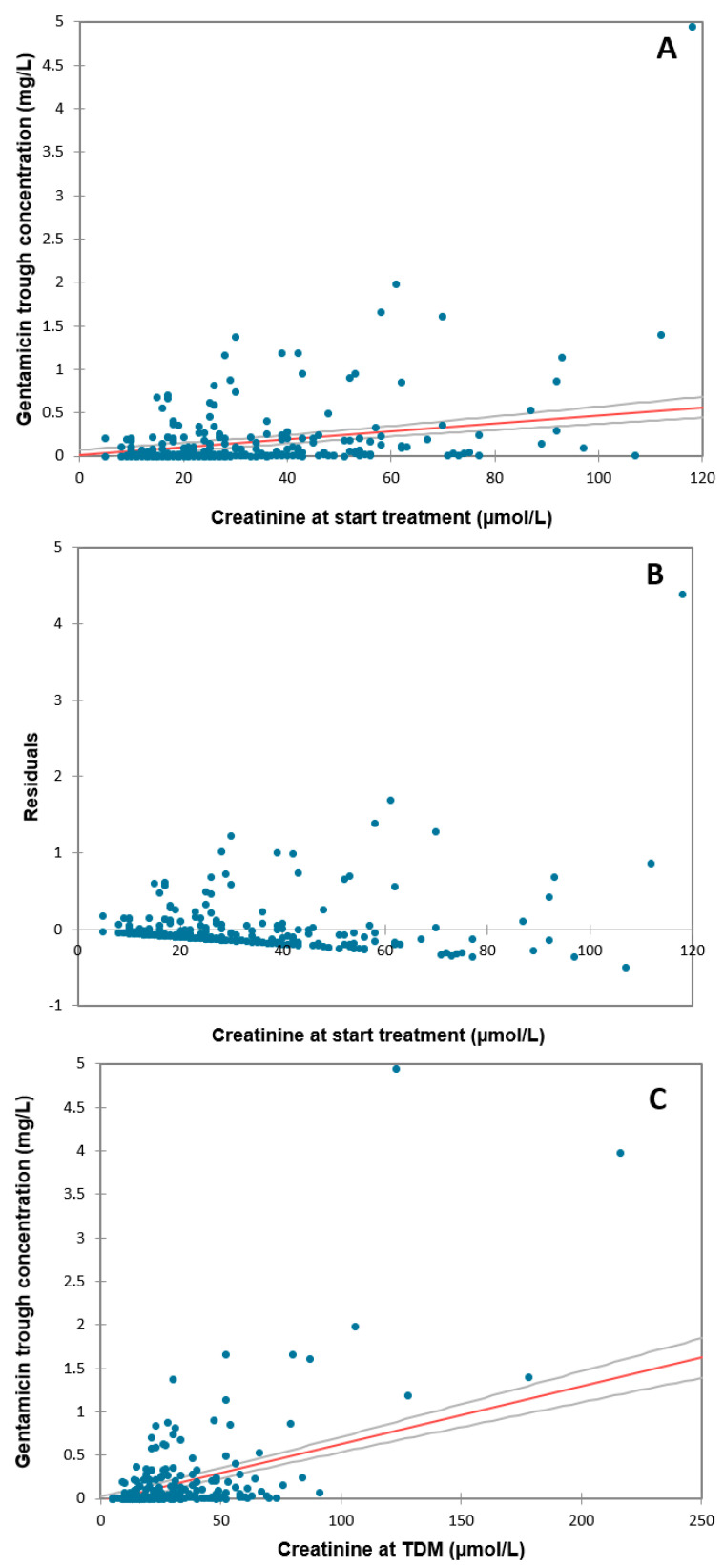
Correlation of gentamicin trough concentrations with creatinine concentrations at the start of antibiotic treatment and at TDM. Deming regression of antibiotic trough concentrations with creatinine concentrations taken within 24 h before the start of gentamicin treatment (**A**), with residuals analysis (**B**); and within 24 h before the TDM sample (**C**), with residuals analysis (**D**). Circles represent single-patient creatinine and gentamicin trough concentrations. The solid line represents the linear regression line.

**Table 1 antibiotics-12-00810-t001:** Overview of dose advice for gentamicin used within the Beatrix Children’s Hospital and the Dutch Pediatric Formulary.

Cohort	BKZ		DPF	
	Daily Dose (mg/kg)	Dose Interval (hours)	Daily Dose (mg/kg)	Dose Interval (hours)
**Neonates**				
GA < 32 weeks, <7 days PNA	5	48	5	48
GA < 32 weeks, ≥7 days PNA	5	48	4	24
GA 32–37 weeks, <7 days PNA	5	36	5	36
GA 32–37 weeks, ≥7 days PNA	5	36	4	24
GA > 37 weeks	5	24	4	24
**Children**				
1 month–12 years	7	24	7	24
**Adolescents**				
12–18 years *(before 2021)*	7	24	7	24
12–18 years *(after 2021)*	5	24	5	24

BKZ: Beatrix Children’s Hospital, DPF: Dutch Pediatric Formulary, GA: gestational age at birth, PNA: postnatal age.

**Table 2 antibiotics-12-00810-t002:** Clinical characteristics and gentamicin concentrations of the neonates and children included.

Characteristics	Neonates	Children
Number of subjects	335	323
Number of cycles	349	358
PNA at treatment start	1 day (0–7)	2.21 years (0.51–9.69)
PNA < 7 days (*n* of cycles)	260	-
PNA ≥ 7 days (*n* of cycles)	89	-
GA at birth (weeks)	38 (33–40)	-
GA < 32 weeks (*n* of cycles)	80	-
GA 32–37 weeks (*n* of cycles	66	-
GA ≥ 37 weeks (*n* of cycles)	203	-
1 month–12 years (*n* of cycles)	-	286
12–18 years (*n* of cycles)	-	72
Weight (kg)	2.79 (1.75–3.60)	12.8 (6.99–31.65)
Male/female	62/38	55/45
Creatinine concentration at start of treatment (µmol/L)	-	25 (17–40), *n* = 258
Creatinine concentration at TDM (µmol/L)	-	24 (16–40), *n* = 236
Gentamicin dose (mg/kg)	5.00 (5.00–5.02)	7.00 (6.96–7.02)
Gentamicin trough concentration (mg/L)	0.92 (0.40–1.44)*^n^*^=312^	0.04 (0.00–0.22)*^n^*^=334^
Gentamicin peak concentration (mg/L)	11.14 (9.09–12.58)*^n^*^=274^	14.77 (12.43–17.80)*^n^*^=317^

Numbers are presented as median (IQR) or *n* = number of cycles. Gentamicin concentrations were measured in serum. PNA: postnatal age, GA: gestational age, TDM: therapeutic drug monitoring.

**Table 3 antibiotics-12-00810-t003:** Overview of the safety and efficacy target attainment of gentamicin in each cohort (supported by Figure 2).

**Safety (Trough Concentration)**		**Safe (%)**	**Toxic (%)**
**Neonates total (*n* = 312)**		53.8	46.2
GA < 32 weeks (5 mg/kg per 48 h) (*n* = 76)		68.4	31.6
PNA < 7 days (*n* = 63)		63.5	36.5
PNA ≥ 7 days (*n* = 13)		92.3	7.7
GA 32–37 weeks (5 mg/kg per 36 h) (*n* = 53)		66.0	34.0
PNA < 7 days (*n* = 41)		56.1	43.9
PNA ≥ 7 days (*n* = 12)		100.0	0.0
GA > 37 weeks (5 mg/kg per 24 h) (*n* = 149)		34.9	65.1
GA > 37 weeks (4 mg/kg per 24 h) (*n* = 34)		85.3	14.7
**Children total (*n* = 334)**		90.1	9.9
1 month-12 years (7 mg/kg per 24 h) (*n* = 266)		89.8	10.2
12–18 years (5 mg/kg per 24 h) (*n* = 25)		100.0	0.0
12–18 years (7 mg/kg per 24 h) (*n* = 43)		86.0	14.0
**Efficacy (Peak Concentration)**	**Subtherapeutic (%)**	**Therapeutic (%)**	**Supratherapeutic (%)**
**Neonates total (*n* = 274)**	10.6	54.0	35.4
GA < 32 weeks (5 mg/kg per 48 h) (*n* = 62)	8.1	50.0	41.9
PNA < 7 days (*n* = 50)	4.0	46.0	50.0
PNA ≥ 7 days (*n* = 12)	25.0	66.7	8.3
GA 32–37 weeks (5 mg/kg per 36 h) (*n* = 49)	4.1	57.1	38.8
PNA < 7 days (*n* = 37)	0.0	48.6	51.4
PNA ≥ 7 days (*n* = 12)	16.7	83.3	0.0
GA > 37 weeks (5 mg/kg per 24 h) (*n* = 130)	6.2	54.6	39.2
GA > 37 weeks (4 mg/kg per 24 h) (*n* = 33)	42.4	54.5	3.0
**Children total (*n* = 317)**	51.7	31.2	17.0
1 month-12 years (7 mg/kg per 24 h) (*n* = 254)	53.5	33.1	13.4
12–18 years (5 mg/kg per 24 h) (*n* = 27)	63.0	25.9	11.1
12–18 years (7 mg/kg per 24 h) (*n* = 36)	30.6	22.2	47.2

Safety was defined as trough concentration <1 mg/L for neonates and <0.5 mg/L for children. *N* = number of cycles included in the analysis. Therapeutic concentrations were defined as peak concentration: 8–12 mg/L for neonates and 15–20 mg/L for children. GA: gestational age, PNA: postnatal age.

## Data Availability

The data presented in this study are available on request from the corresponding author. The data are not publicly available due to privacy/ethical restrictions.

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
