# Peer review of "Evaluation of Dosing Guidelines for Gentamicin in Neonates and Children"

_antibiotics, 2023, doi:10.3390/antibiotics12050810_

Round 1

Reviewer 1 Report

This is an interesting topic and the manuscript is very well-written. Retrospective single centre cohort study. Methods are explained in detail and results provide valuable insight into target attainment of gentamicin among neonates and children according to dosing regimens in current guidelines. About 50% of the drug concentrations are subtherapeutic or supratherapeutic, which emphasize the conclusion that further steps and real-world studies are needed to ensure target attainment of gentamicin in neonates and children. The authors give also many practical recommendations regarding the suitability of gentamicin dosing in current guidelines. 

Author Response

Reviewer 1

This is an interesting topic and the manuscript is very well-written. Retrospective single centre cohort study. Methods are explained in detail and results provide valuable insight into target attainment of gentamicin among neonates and children according to dosing regimens in current guidelines. About 50% of the drug concentrations are subtherapeutic or supratherapeutic, which emphasize the conclusion that further steps and real-world studies are needed to ensure target attainment of gentamicin in neonates and children. The authors give also many practical recommendations regarding the suitability of gentamicin dosing in current guidelines. 

We thank the reviewer for the positive feedback.

Reviewer 2 Report

Hollander EM and van Tuinen EL et al. evaluates the current dosing guidelines in the Netherlands for the aminoglycoside antibiotic gentamicin in neonates and children using retrospective gentamicin concentration data from patients admitted to the Beatrix Children’s Hospital (BKZ).

My comments are as follows:

Major

While the manuscript is well written, in the absence of the treatment response, the method used to answer the question of whether the dosing guidelines are effective, is flawed. Several assumptions have been made about the safe and efficacious target concentrations for Gentamicin and correlated data (multiple measurements from the same patient) have been used to draw conclusions. The retrospective nature of the study limits its utility. Furthermore, the framework used to evaluate impact of covariates is incorrect. A population PK-PD model would have been the appropriate approach to utilize.

Minor comments:

1.      I recommend including information on the pharmacokinetics of Gentamicin (volume of distribution, clearance, half-life, etc.) and the impact of age on half-life in the ‘Introduction’ section. Also include information on the typical duration of Gentamicin treatment

1.      Given the short half life and QD dosing of Gentamicin, steady state should be reached after the first 2-3 days of once daily dosing in most patients. So any dose adjustments to attain the effective concentrations must be made in the first 2-3 days.

As such, I do not think evaluating Gentamicin concentrations for several days of treatment is useful unless any patient specific characteristics change (eg: kidney function). I would recommend presenting the peak and trough concentrations of Gentamicin after the first 3 doses (presented as concentrations after dose 1, 2 and 3) of the once daily dosing regimen instead of the entire treatment range.

2.      Suggest to clarify what is meant by ‘cycle’ of Gentamicin dosing (how many days or doses of Gentamicin?)

3.      Since multiple trough and peak concentrations from the same patient are included, there exists an inherent correlation between these concentrations. Statistical tests (eg: regression) cannot be performed on the entire dataset since these data are correlated. Suggest to either adjust for this correlation or only test 1 representative concentration (eg: mean or median) per patient.

4.      Table 2, suggest to clarify what was the matrix used to measure these Gentamicin concentrations (plasma, serum or blood)?

5.      I recommend presenting the summary statistics (median, IQR) for number of cycles per patient in Table 2. Please present the number of subjects

Author Response

Reviewer 2

Hollander EM and van Tuinen EL et al. evaluates the current dosing guidelines in the Netherlands for the aminoglycoside antibiotic gentamicin in neonates and children using retrospective gentamicin concentration data from patients admitted to the Beatrix Children’s Hospital (BKZ). My comments are as follows:

Major

While the manuscript is well written, in the absence of the treatment response, the method used to answer the question of whether the dosing guidelines are effective, is flawed. Several assumptions have been made about the safe and efficacious target concentrations for Gentamicin and correlated data (multiple measurements from the same patient) have been used to draw conclusions. Thank you for this comment. We have extended our analysis by using mixed models to identify the effect of multiple measurement per patient. In the method section, paragraph 4.4 Data analysis we have added in line 529-534 the following: “Differences in gentamicin peak and trough concentrations among the different subcohorts were additionally analyzed using linear mixed model analysis in order to confirm the observed differences in target attainment but then also accounting for the correlated nature of the data (i.e. some individuals having multiple concentration measurements). Here age subgroups were entered as fixed effect while a random effect was allowed for subject ID. Akaike’s information criterion (AIC) was used to prioritize the best fitting covariance structure.” Furthermore, we showed after performing the linear mixed model analysis that multiple measurement from the same patient did not influence our results and conclusion. These results have been added both for neonates in line 176-187. We added the following: “In addition, linear mixed model analysis was performed to confirm the observed differences in gentamicin peak and trough concentrations among the age subgroups (fixed effect) while allowing the presence of multiple concentration measurements for some neonates (random effect for individual subjects). When grouping all neonates together per age subgroup (<32 weeks GA, 32-37 weeks GA, and term neonates), neonates at term showed significantly lower peak concentrations but higher trough concentrations compared to neonates <32 weeks GA (P=0.010 and P<0.001, respectively). When studying differences between <7 and ≥7 days PNA, peak concentrations were significantly higher in neonates <7 days PNA compared to neonates ≥7 days PNA in both the <32 weeks GA and 32-37 subgroups (P<0.001 and P=0.001, respectively). Similarly, trough concentrations were higher in neonates <7 days PNA compared to neonates ≥7 days PNA in both <32 weeks GA and 32-37 weeks subgroups (P=0.005 and P<0.001, respectively)”. For children we have added the results in line 198-203 as follows: “ Linear mixed model analysis, performed to account for the presence of multiple measurements for some children, confirmed these observations, showing on average higher peak concentrations in children >12 years receiving 7 mg/kg compared to children >12 years receiving 5 mg/kg and children <12 years (both P<0.001), and on average lower trough concentrations in children >12 years receiving 5 mg/kg compared to those receiving 7 mg/kg (P=0.018)” .

The retrospective nature of the study limits its utility. Furthermore, the framework used to evaluate impact of covariates is incorrect. A population PK-PD model would have been the appropriate approach to utilize. We agree with the reviewer that developing a population PK/PD would be a good approach to identify relevant covariates and develop evidence-based dosing regimens for gentamicin. However, this is out of the scope of this retrospective study. In addition, comments by the editor of Antibiotics mentioned that it would not be doable to perform a population PKPD study of gentamicin in neonates and children.

Minor comments:

  1. I recommend including information on the pharmacokinetics of Gentamicin (volume of distribution, clearance, half-life, etc.) and the impact of age on half-life in the ‘Introduction’ section. Also include information on the typical duration of Gentamicin treatment. Information on the pharmacokinetics of gentamicin and impact of age in neonates and children has been added in line 48-53.

  1. Given the short half life and QD dosing of Gentamicin, steady state should be reached after the first 2-3 days of once daily dosing in most patients. So any dose adjustments to attain the effective concentrations must be made in the first 2-3 days. We agree with the reviewer and confirm that this is our clinical practice.

As such, I do not think evaluating Gentamicin concentrations for several days of treatment is useful unless any patient specific characteristics change (eg: kidney function). I would recommend presenting the peak and trough concentrations of Gentamicin after the first 3 doses (presented as concentrations after dose 1, 2 and 3) of the once daily dosing regimen instead of the entire treatment range.  We do not agree with the reviewer. The reviewer assumes that septic/infected patients have stable pharmacokinetics, which mostly is not the case. Patient receiving gentamicin have serious infections and these are accompanied by changes in pharmacokinetics, notably volume of distribution and many times also clearance. Fluid balance changes during treatment and pharmacokinetics change. Therefore, we included also the subsequent gentamicin levels. Further, by analysing the data using linear mixed modelanalysis, we corrected for multiple sampling in some patients.

  1. Suggest to clarify what is meant by ‘cycle’ of Gentamicin dosing (how many days or doses of Gentamicin?) A cycle involves a treatment of at least 1 dose of gentamicin, we added this in line 375.
  2. Since multiple trough and peak concentrations from the same patient are included, there exists an inherent correlation between these concentrations. Statistical tests (eg: regression) cannot be performed on the entire dataset since these data are correlated. Suggest to either adjust for this correlation or only test 1 representative concentration (eg: mean or median) per patient. We have extended our analysis by using mixed models to identify the effect of multiple measurement per patient. In the method section, paragraph 4.4 Data analysis we have added in line 529-534 the following: “Differences in gentamicin peak and trough concentrations among the different subcohorts were additionally analyzed using linear mixed model analysis in order to confirm the observed differences in target attainment but then also accounting for the correlated nature of the data (i.e. some individuals having multiple concentration measurements). Here age subgroups were entered as fixed effect while a random effect was allowed for subject ID. Akaike’s information criterion (AIC) was used to prioritize the best fitting covariance structure.” Furthermore we showed after performing the linear mixed model analysis that multiple measurment from the same patient did not influence our results and conclusion. Those results have been added both for neonates in line 176-187. We described the following: “In addition, linear mixed model analysis was performed to confirm the observed differences in gentamicin peak and trough concentrations among the age subgroups (fixed effect) while allowing the presence of multiple concentrations measurements for some neonates (random effect for individual subjects). When grouping all neonates together per age subgroup (<32 weeks GA, 32-37 weeks GA, and term neonates), neonates at term showed significantly lower peak concentrations but higher trough concentrations compared to neonates <32 weeks GA (P=0.010 and P<0.001, respectively). When studying differences between <7 and ≥7 days PNA, peak concentrations were significantly higher in neonates <7 days PNA compared to neonates ≥7 days PNA in both the <32 weeks GA and 32-37 subgroups (P<0.001 and P=0.001, respectively). Similarly, trough concentrations were higher in neonates <7 days PNA compared to neonates ≥7 days PNA in both <32 weeks GA and 32-37 weeks subgroups (P=0.005 and P<0.001, respectively)”. For children we have added those information in line 198-203 as follows: “ Linear mixed model analysis, performed to account for the presence of multiple measurements for some children, confirmed these observations, showing on average higher peak concentrations in children >12 years receiving 7 mg/kg compared to children >12 years receiving 5 mg/kg and children <12 years (both P<0.001), and on average lower trough concentrations in children >12 years receiving 5 mg/kg compared to those receiving 7 mg/kg (P=0.018)” .
  3. Table 2, suggest to clarify what was the matrix used to measure these Gentamicin concentrations (plasma, serum or blood)?

The matrix used to measure gentamicin concentrations was serum. We added this to the text below table 2.

  1. I recommend presenting the summary statistics (median, IQR) for number of cycles per patient in Table 2. Please present the number of subjects

We added the number of subjects to table 2.

Reviewer 3 Report

Comments for Authors;

The article entitled as “Evaluation of dosing guidelines for gentamicin in neonates and  children” by Esther M. Hollander ET al .

Some of comments for the authors may please needs clarification:

In abstract section line 25 please add the name of hospital setting where you conducted this study.

Rephrase line 26 of the abstract section.

Figure 1 of the results section may please be converted in tabular form for the readers easy understandability.

Please add the limitations of yours study also the future perspective of yours study as recommendation section.

I think in my opinion the supplementary file may not be attached in main text of the manuscript.

Recheck the grammar and remove the typo mistakes.

Author Response

The article entitled as “Evaluation of dosing guidelines for gentamicin in neonates and  children” by Esther M. Hollander ET al. Some of comments for the authors may please needs clarification:

In abstract section line 25 please add the name of hospital setting where you conducted this study.

We added the name of the hospital setting where this study was conducted.

Rephrase line 26 of the abstract section.

We rephrased line 26.

Figure 1 of the results section may please be converted in tabular form for the readers easy understandability. We agree with the reviewer that this is indeed a good suggestion. However, to visualize the flow of the data selection a figure provides a better insight in this process than a table.

Please add the limitations of yours study also the future perspective of yours study as recommendation section.
Thank you for this comment. The limitations of our study are mentioned in line 327-347. And future perspectives of our study are mentioned in line 308-325. We have also added the subheadings “ recommendations” and “ limitations”  above the paragraphs.

I think in my opinion the supplementary file may not be attached in main text of the manuscript. We agree with the reviewer supplementary file with Table 1 will not be in the main text but reported as a supplementary appendix.

Recheck the grammar and remove the typo mistakes.

We checked the grammar and removed all the typo mistakes we could find.

Reviewer 4 Report

I have evaluated the manuscript (Antibiotics--2333308) titled “Evaluation of dosing guidelines for gentamicin in neonates and children” by Mian et. al. and the authors evaluated retrospectively cohort study of dosing guidelines for antibiotic gentamicin in neonates and children between January 2019 and July 2022 at Beatrix Children's Hospital. I found this article interesting for the readers and followed the journal Antibiotics’ scope. The overall presentation and discussion of this manuscript is excellent with table and figures.

I would recommend this article be published in Antibiotics after minor corrections. 

The author needs to address the following comments/corrections.

 1.     The author should correct the format of references wherever needed (e.g Year Bold, Volume Italic etc.).

2.     Use either hours or h to represent time.

3.     In Fig 1 and table 1 total number of cycles shown 707, however, in the line 81, it is 737 cycles (735 gentamicin treatment cycles).

4.     As the study includes the Covid-19 period: the author could have comments on any effects on Covid-19 in this study.

5.     The author could move supplementary table 1 in the discuss as there is only one table.

Author Response

I have evaluated the manuscript (Antibiotics--2333308) titled “Evaluation of dosing guidelines for gentamicin in neonates and children” by Mian et. al. and the authors evaluated retrospectively cohort study of dosing guidelines for antibiotic gentamicin in neonates and children between January 2019 and July 2022 at Beatrix Children's Hospital. I found this article interesting for the readers and followed the journal Antibiotics’ scope. The overall presentation and discussion of this manuscript is excellent with table and figures. I would recommend this article be published in Antibiotics after minor corrections. 

The author needs to address the following comments/corrections.

  1. The author should correct the format of references wherever needed (e.g Year Bold, Volume Italic).

We thank the reviewer for this comment and adapted everything according to the style of the journal.

  1. Use either hours or h to represent time.

We adjusted every ‘h’ to ‘hours’.

  1. In Fig 1 and table 1 total number of cycles shown 707, however, in the line 81, it is 737 cycles (735 gentamicin treatment cycles).

The reviewer is right, we adjusted the numbers in line 81.

  1. As the study includes the Covid-19 period: the author could have comments on any effects on Covid-19 in this study.
    The study indeed includes the Covid-19 period. However, we do not think covid-19 had any relevant effects on this study. Therefore, we did not mention covid-19.
  2. The author could move supplementary table 1 in the discuss as there is only one table. We received a contradictory suggestion from reviewer 3. Therefore, we decided to keep the supplementary table 1 as supplementary file.

Round 2

Reviewer 2 Report

Thank you for addressing my comments.